# Efficacy of High-Definition Transcranial Alternating Current Stimulation (HD-tACS) at the M1 Hotspot Versus C3 Site in Modulating Corticospinal Tract Excitability

**DOI:** 10.3390/biomedicines12112635

**Published:** 2024-11-19

**Authors:** Hao Meng, Michael Houston, Nicholas Dias, Chen Guo, Gerard Francisco, Yingchun Zhang, Sheng Li

**Affiliations:** 1Department of Physical Medicine and Rehabilitation, McGovern Medical School, University of Texas Health Science Center at Houston, Houston, TX 77030, USA; hao.meng@uth.tmc.edu (H.M.); gerard.e.francisco@uth.tmc.edu (G.F.); 2Department of Biomedical Engineering, University of Houston, Houston, TX 77204, USA; m.jameshouston81@gmail.com (M.H.); nicholascdias@gmail.com (N.D.); cxg1495@miami.edu (C.G.); y.zhang@miami.edu (Y.Z.); 3Desai Sethi Urology Institute, University of Miami, Miami, FL 33136, USA; 4Department of Biomedical Engineering, University of Miami, Miami, FL 33146, USA; 5TIRR Memorial Hermann Hospital, Houston, TX 77030, USA; 6Miami Project to Cure Paralysis, University of Miami, Miami, FL 33136, USA

**Keywords:** transcranial alternating current stimulation, action potentials, cortical excitability, motor cortex

## Abstract

Previous studies have shown that beta-band transcranial alternating current stimulation (tACS) applied at the M1 hotspot can modulate corticospinal excitability. However, it remains controversial whether tACS can influence motor unit activities at the spinal cord level. This study aims to compare the efficacy of applying tACS over the hotspot versus the conventional C3 site on motor unit activities and subsequent behavioral changes. This study used a randomized crossover trial design, where fifteen healthy participants performed a paced ball-squeezing exercise while receiving high-definition tACS (HD-tACS) at 21 Hz and 2 mA for 20 min. HD-tACS targeted either the flexor digitorum superficialis (FDS) hotspot or the C3 site, with the order of stimulation randomized for each participant and a 1-week washout period between sessions. Motor unit activities were recorded from the FDS. HD-tACS intervention significantly reduced the variability of motor unit firing rates and increased force variability during isometric force production. The significant modulation effects were seen only when the intervention was applied at the hotspot, but not at the C3 site. Our findings demonstrate that HD-tACS significantly modulates motor unit activities and force variability. The results indicate that cortical-level entrainment by tACS can lead to the modulation of spinal motor neuron activities. Additionally, this study provides further evidence that the C3 site may not be the optimal target for tACS intervention for hand muscles, highlighting the need for personalized neuromodulation strategies.

## 1. Introduction

Transcranial alternating current stimulation (tACS) is a promising non-invasive brain stimulation method that delivers low-intensity electrical currents to the scalp, modulating intracranial neuronal oscillation [1]. tACS modulates neural activity by entraining neurons and synchronizing their firings at a specific frequency [2,3,4]. tACS can modulate the excitability of cortical neurons by inducing rhythmic shifts in their membrane potential, making them alternately more prone to firing (depolarization) or less prone to firing (hyperpolarization) [3]. A previous review article showed that, while beta-band tACS can modulate the primary motor cortex (M1) excitability, using intensities above 1 mA resulted in a more robust modulation effect, with the tACS montages also influencing these effects [5]. Our recent review article suggests tACS has potential in modulating motor cortex excitability compared to other techniques but highlights that the modulation effects can be affected by various environments, such as the montage setup, intensity, frequency, and duration [4].

In the conventional transcranial electrical stimulation (tES) setup, one electrode is usually placed on the M1, with the reference electrode placed on the contralateral supraorbital lobe. However, high-definition tACS (HD-tACS) is a newly modified tES technique that improves the focality of conventional tES. In a standard 4 × 1 high-definition setup, the central electrode is placed on the targeted cortex, with four reference electrodes surrounding the central electrode [6]. Previous studies have shown evidence that high-definition transcranial direct stimulation (HD-tDCS) modulates cortical excitability more effectively than conventional tDCS [7,8]. Additionally, Heise, Kortzorg [9] have shown that employing HD-tACS at the M1 hotspot can significantly enhance cortical excitability, as measured by motor evoked potentials (MEPs), both during and after stimulation, compared to the conventional electrode montage placement.

Corticospinal input refers to the signals that originate in the cerebral cortex and are transmitted down the spinal cord via the corticospinal tract [10,11,12]. Corticospinal axons travel from the sensorimotor cortex through the internal capsule, cerebral peduncles, pons, and medulla, ultimately synapsing onto the lower motor neurons of the spinal cord [13]. Therefore, corticospinal excitability, as manifested by MEP assessment, is subject to changes in the cortical and/or spinal regions [14]. In our previous study, we found that beta-band HD-tACS applied at the M1 hand hotspot increased post-stimulation MEP amplitudes of flexor digitorum superficialis (FDS) muscles [15]. Similar modulation effects were also observed in other studies [16,17,18,19]. However, if entrainment occurs after tACS, it is reasonable to anticipate its effects should be transmitted via the corticospinal tract to the motor neurons that innervate the corresponding muscles as well. Therefore, examining the motor unit activity in muscles should provide meaningful evidence that cortical-level entrainment by tACS can lead to modulation of spinal motor neurons. Interestingly, despite computer model studies showing positive entrainment effects in the spinal motor neurons after tACS, its modulation effect was not clearly observed in human subjects [20].

Thus, this study has two primary aims: (1) to determine whether HD-tACS can modulate spinal motor neuron behaviors through the corticospinal tract using a more focal montage setup and at a higher intensity, and (2) whether HD-tACS at the identified hotspot for the upper limb can modulate the behaviors differently from the conventionally used C3 site. Our recent data have shown that HD-tACS applied over the M1 hand hotspot can modulate cortical excitability, as reflected by increased peak-to-peak MEPs [15]. It is reasonable to anticipate comparable modulation effects at the spinal lower neuron level. Our first hypothesis is that a short-term HD-tACS intervention applied over M1 can significantly modulate motor unit activities, as reflected by reduced variability of the firing rate. Our second hypothesis is that an HD-tACS intervention applied at the hotspot can significantly modulate motor unit behaviors, including its behavioral consequences in force production.

## 2. Materials and Methods

Participants:

A total of 15 healthy right-handed participants participated in this cross-over study (5 females, 29.8 ± 11.9 years of age; 10 males, 34.6 ± 11.2 years of age). The study protocol was approved by the Committee for the Protection of Human Participants (CPHS) of the University of Texas Health Science Center at Houston and was conducted in adherence to the standards established by the Declaration of Helsinki. Prior to the commencement of the study, written informed consent was obtained from all participants.

Protocols:

This study employed a randomized crossover trial design. Participants were fitted with a suitable 10–20 electroencephalogram (EEG) cap and seated comfortably, allowing their arms and hands to rest naturally. High-density surface electromyography (HD-sEMG) electrodes were positioned over the muscle belly of the contralateral flexor digitorum superficialis (FDS) muscle using a 64-channel, two-dimensional electrode array (TMSi, Oldenzaal, The Netherlands). The High-Density Surface Electromyography (HD-sEMG) signals were recorded with a Twente Medical Systems International SAGA amplifier (TMSi, Oldenzaal, The Netherlands) at a sampling frequency of 4 kHz. The data were filtered at 10–500 Hz with 4th order Butterworth Filter + notch filter at 60 Hz.

Four force transducers (PCB Piezotronics Inc., Depew, NY, USA) were positioned in front of the participant’s dominant hand, with spacing adjusted to comfortably accommodate the index, middle, ring, and little fingers. Participants were given sufficient time to adjust the positions of the transducers for comfort before the actual test. The sampling frequency was set at 1000 Hz.

The M1 FDS hotspot was determined using single-pulse transcranial magnetic stimulation (TMS) with a figure-of-8 coil connected to a Magstim Bistim2 200 stimulator (Magstim Company, Whitland, Wales, UK). The TMS coil was positioned tangentially over the scalp, with its handle directed backward at a 45-degree angle. The M1 FDS hotspot was identified as the location that consistently elicited the highest peak-to-peak MEPs from the FDS. This hotspot was marked with a permanent pen, and its location was documented to ensure consistent targeting between visits.

Participants were required to perform both maximum voluntary contraction (MVC) and isometric contraction tests before and after the HD-tACS intervention for each visit. Prior to each test, participants were instructed to rest the fingers of their dominant hand naturally on the force transducers, ensuring no pressure was applied. They were also instructed to avoid moving the transducers or lifting their wrists or arms during the test.

In the MVC test, participants were instructed to press on the transducers with their four fingers as hard as they could for 5 s. The highest cumulated force value from three MVC attempts was selected as the MVC value. The isometric contraction test consisted of a 10 s force ramp-up phase followed by a 40 s contraction phase targeting 10% of the maximum voluntary contraction (MVC). Visual targets were displayed on the screen, which were programmed using LabVIEW (National Instruments, Austin, TX, USA). During the ramp-up phase, participants gradually increased the cumulative force until the on-screen trajectory met the preset 10% MVC. In the subsequent contraction phase, participants were instructed to maintain the cumulative force trajectory as close to the 10% MVC target line as possible. Participants were given 2 to 3 practice trials to familiarize themselves with the task before the experiment began. No HD-tACS intervention was administered during either test.

HD-tACS intervention was administered using the Soterix MxN HD-tES system (Soterix Medical Inc., Woodbridge, NJ, USA). The central electrode was placed on the pre-marked M1 FDS hotspot, and four reference electrodes were evenly positioned in the surrounding areas in accordance with the spacing guidelines of the 10–20 system (Figure 1). The positions of the electrodes were documented to ensure consistency between visits. Each participant received two HD-tACS interventions with a 1-week washout period. The HD-tACS treatment was administered at either the hotspot or the C3 site, based on random assignment. The duration of each HD-tACS intervention was 20 min, with the intensity set at 2 mA and the frequency at 21 Hz. During the HD-tACS intervention, participants were instructed to gently squeeze a small stress ball at a pace of 15 squeezes per minute until the end of the intervention.

Outcome Measures:

The procedures for MUAP decomposition are detailed in our previous work [21,22]. Identical spike trains were identified by directly comparing each identified component. If more than 40% of the identified peaks were shared between a pair of spike trains, the component with the lower inter-spike interval (ISI) coefficient of variation (CoV) was retained. MUAP spatial potential distributions were determined by solving a least-squares problem to minimize the difference between the measured HD-sEMG signal and the estimated signal generated by the convolved spike trains and their corresponding MUAP distributions. Hierarchical clustering was used to cluster the amplitude distributions of MUAP shapes derived from each spike train. Within each cluster, the identified spike train with the smallest inter-spike interval (ISI) coefficient of variation (CoV) was retained. Finally, components with firing rates outside the physiologically appropriate frequency range (4–50 Hz) were not considered for further analysis. ISIs greater than 0.5 s were excluded from the calculation of the firing rates to avoid influence from components that were recruited late or started and stopped intermittently.

MUAP characteristics considered for analysis included the total number of decomposed MUAPs (MUAP_Num), amplitude of MUAPs (MUAP_Amp), firing rate of MUAPs (MUAP_FR), and the coefficient of variation of MUAP firing rates (CoV_MUAP_FR). MUAP amplitude was defined as the maximal root mean square (RMS) amplitude across all channels, based on a representative waveform composed of the average of all firing instances. The average firing rate of MUAPs was calculated by first determining the instantaneous firing rate through the convolution of the binary spike trains with a one-second window. Subsequently, a one-second Gaussian filter was applied to smooth the instantaneous firing rate values. Finally, these smoothed firing rate values were averaged across the entire trial to obtain a representative average MUAP firing rate. The coefficient of variation of the MUAP firing rate (CoV_MUAP_FR) was calculated based on this average MUAP firing rate.

The variability of force was assessed by calculating the CoV of force (CoV_Force), which was the ratio of the standard deviation (SD) of the total force to its mean for the total 40 s, multiplied by 100. Here, the total force was the sum of the forces produced by each individual finger. All computations were performed using MATLAB 2022b (MathWorks, Natick, MA, USA).

Statistical Analysis:

The normality of distributions was assessed using the Shapiro–Wilk test, and the homogeneity of variances was evaluated using Levene’s test and Mauchly’s test for sphericity. When the assumption of sphericity was found to be violated, the Greenhouse–Geisser correction was applied. Two-way repeated measures Analysis of Variances (ANOVAS) were performed to determine the effect of Stimulation_Site (Hotspot vs. C3) and Time (Pre vs. Post) on MUAP_Num, MUAP_Amp, MUAP_FR, CoV_MUAP_FR and CoV_Force. Post hoc tests with Bonferroni corrections were conducted only when significant main effects were observed. All statistical analyses were performed using SPSS 27 (IBM Corp., Armonk, NY, USA), with the significance level set at *p* < 0.05.

## 3. Results

A total of 15 healthy right-handed participants participated in this study (five females, 29.8 ± 11.9 years of age; ten males, 34.6 ± 11.2 years of age). Overall, the application of HD-tACS significantly influenced two key measures: the variability of motor unit firing rate (CoV_MUAP_FR) and the variability of the force produced by the participants (CoV_Force). Other characteristics related to the motor unit action potential (MUAP) were not significantly affected (Table 1).

For CoV_MUAP_FR (the measure of how consistently the muscle’s motor units fire), there was a significant effect over time, meaning that HD-tACS caused changes in the consistency of motor unit firing as the session progressed (F_(1,28)_ = 5.012, *p* = 0.033, η^2^ = 0.152, Figure 2a). However, there were no significant differences based on the stimulation site (F_(1,28)_ = 0.184, *p* = 0.671, η^2^ = 0.007) or in the interaction between time and stimulation site (F_(1,28)_ = 1.171, *p* = 0.288, η^2^ = 0.04). A post hoc analysis showed that these changes were only significant when HD-tACS was applied to the FDS hotspot. Specifically, there was a 12.92% reduction in the variability of the motor unit firing rate after HD-tACS was applied to the FDS hotspot (*p* = 0.026, Figure 3a), but no significant change when stimulation was applied to the C3 site (*p* = 0.42).

For CoV_Force (the measure of how consistently the participants generated force), there was also a significant effect over time, indicating that the stimulation affected how consistently force was applied throughout the task (F_(1,28)_ = 5.611, *p* = 0.025, η^2^ = 0.167, Figure 2b). However, there were no significant differences based on the stimulation site (F_(1,28)_ = 0.051, *p* = 0.822, η^2^ = 0.002) or interaction effects (F_(1,28)_ = 0.286, *p* = 0.597, η^2^ = 0.01) effects. A post hoc analysis revealed that the variability of force output increased by 47.37% after HD-tACS was applied to the FDS hotspot (*p* = 0.049, Figure 3b), but no significant changes were found at the C3 site.

However, the number of identified MUs was not significantly altered, as no main effects were identified for Location (F_(1,28)_ = 0.49, *p* = 0.487) or Time (F_(1,28)_ = 0.03, *p* = 0.862). Meanwhile, there was a main effect of Location (F_(1,28)_ = 5.79, *p* = 0.021), but not Time (F_(1,28)_ = 0.8, *p* = 0.375) for MUAP amplitude. However, post hoc analysis did not reveal any significant changes in MUAP amplitude when stimulating the hotspot (*p* = 0.532) or C3 site (*p* = 0.216). As for the average firing rates of MUAPs, no significant effects for Location (F_(1,28)_ = 1.24, *p* = 0.272) or Time (F_(1,28)_ = 0.79, *p* = 0.380) were found.

## 4. Discussion

The results of our study revealed two key findings. First, HD-tACS significantly decreased the variability of the MUAP firing rate and increased the variability of cumulative isometric contraction force; second, the significant modulation effects were observed only when HD-tACS was applied to the FDS hotspot, but not to the C3 site.

tACS is believed to modulate brain activity through two mechanisms: immediate effects during stimulation (online) and persistent effects after stimulation (offline). Both involve entraining brain rhythms to the external stimulation and altering long-range communication patterns between brain areas [3,23]. Additionally, the effects of tACS can extend to large-scale excitability through a few possible mechanisms, including stochastic resonance, rhythm resonance, temporal biasing of spikes, network entrainment, and imposed patterning [1]. Although studies examining the efficacy of HD-tACS in modulating motor cortex or corticospinal excitability are limited, one recent study has demonstrated that HD-tACS has the potential for modulating cortical excitability [6]. In addition, a few studies compared the effects of both HD-tDCS and conventional tDCS, reporting that they modulated cortical activities distinctively [8,12]. The results of the current study are also consistent with a recent study that reported HD-tACS has the potential to modulate cortical excitability [15]. As discussed in the Introduction, our findings of reduced variability in motor unit firing rate provide further evidence and extend previous findings that tACS can modulate corticospinal excitability at the spinal level.

Even though no main effect of Location was reported, additional analysis indicated that the reduction in firing rate variability and the increase in force variability were likely driven by the modulation applied over the hotspot, rather than the C3 site. Specifically, variability in firing rate was reduced by approximately 13%, and variability in force increased by 47% when stimulation was applied over the hotspot, compared to only a 5% reduction and a 26% increase, respectively, at the C3 site. In fact, a recent similar study failed to report any significant modulation effect on MUAP activities when different tACS frequencies were applied over the C3 in healthy participants [20]. One of the critical factors that may lead to the distinct outcome is the different stimulation sites used. Kim, Kim [24] compared the anatomical hand knob area (determined by navigated-TMS) and hand motor hotspot (single-pulse TMS) and found that the hotspot yielded better modulation effects from repetitive transcranial magnetic stimulation (rTMS). Similarly, another recent study showed that the M1 first dorsal interosseous (FDI) hotspot showed better modulation effects than C3, measured by the changes in MEPs [25]. The results of the current study provide additional evidence supporting the importance of precise localization of the target area in improving neuromodulation effects. Furthermore, the current study set the stimulation intensity at 2 mA, rather than the 1 mA used in the study mentioned above. Although previous studies have shown that weak stimulation currents can modulate cortical excitability [26,27], a recent article highlighted that an intensity over 1 mA can achieve a more robust effect than an intensity below 1 mA [5].

However, neither MUAP number nor MUAP amplitude showed changes after the intervention. The oscillating shifts in membrane potential induced by tACS are thought to be strong enough to modulate neuronal firing activities, but not to directly cause them to fire. Instead, this sub-threshold stimulation acts as a timing mechanism, controlling when neurons fire in a manner that is specific to both the frequency of the stimulation and the location of the neurons [28,29,30]. Given the fact that tACS primarily synchronizes the neuron firing activity rather than significantly recruiting more motor units during isometric contraction, the results appear to align with this typically accepted mechanism of tACS.

Motor output variability refers to the unintentional differences in the results of voluntary muscle contractions [31]. Examining the variability of the motor output provides valuable insights into motor performance and the recruitment levels of the corticospinal tract and motor units [32,33]. Although the variability of the MUAP firing rate has not been widely used, it is a key factor in force variability during isometric contraction [34]. The results of the current study demonstrated that the variability of the MUAP firing rate decreased on average by approximately 9% after 20 min of tACS treatment, suggesting that the firing of motor units became more synchronized. However, the variability of isometric contraction force increased by approximately 36%. This seems to be counter-intuitive at the first look. It does actually reflect the underlying mechanisms of neuromuscular force control. Before HD-tACS treatment, motor units fire and oscillate at certain frequencies with individual peaks and troughs. When the firing of motor units becomes less variable after HD-tACS treatment, more oscillation peaks of motor unit firing are aligned or become closer. Such synchrony of the oscillation peaks increases the force output [33]. Similarly, the synchrony of oscillation troughs decreases the force output. In other words, the increased variability of total finger force is a behavioral reflection of the decreased variability of motor unit firing. Given that there were no changes in the number of firing motor units and their amplitude and firing rate, the total force remains unchanged. This phenomenon is visually displayed in Figure 1.

## 5. Limitations

This study presents several advantages over the existing body of literature, as well as a few disadvantages. It benefits significantly from the use of HD-tACS rather than conventional tACS, enabling the more precise delivery of exogenous currents, thus leading to site-specific effects. However, the limitations of this study cannot be disregarded. Beta-band tACS was chosen as the primary stimulation frequency in this study because beta-band oscillations play critical roles in supporting cognitive and motor functions. Nonetheless, previous studies using different frequencies have also observed modulation effects, and one study suggested that an individualized frequency approach may yield better outcomes [16,35,36]. Therefore, further research could focus on optimizing the stimulation frequencies to enhance the efficacy of tACS. Moreover, a sham condition could be included in future studies. C3 is a site that is commonly used for tACS or tDCS. The result of no neuromodulatory effects from active HD-tACS at C3 suggests that it is not likely to observe a placebo effect in this study. Since there were no effects of HD-tACS at C3 on the corticospinal excitability in our recent study [15], this condition could be served as a sham condition. Finally, a limitation of this study is the absence of double-blinding. This lack of blinding introduces the potential for bias, particularly from the investigators, which could influence the application of the intervention.

## 6. Conclusions

The results of the current study demonstrate that beta-band HD-tACS significantly synchronizes motor neuron firing and increases the variability of isometric finger force production, indicating that tACS can neuromodulate spinal motor neuron activities. This study provides additional evidence that the conventionally used C3 site may not be the optimal target for tACS.

## Figures and Tables

**Figure 1 biomedicines-12-02635-f001:**
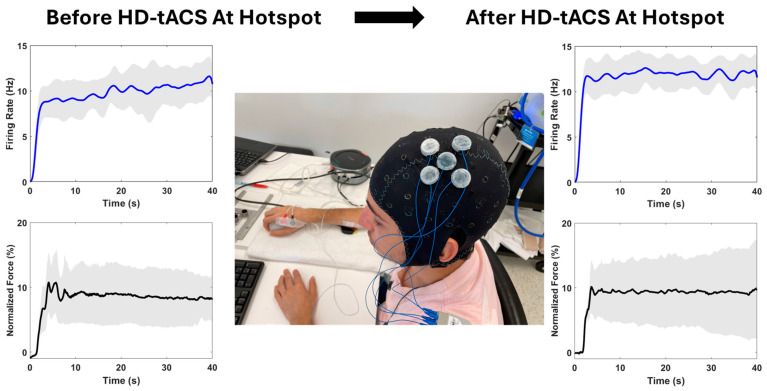
Setup of HD-tACS intervention at the FDS hotspot and representative figures illustrating HD-tACS modulation of force and motor unit action potential (MUAP) firing rate variabilities from a representative participant. Blue and black traces are the mean of all MUAP firing rates and baseline-corrected forces in newtons across time, respectively. Gray shadings indicate the standard deviations across all MUAP firing rates and forces in newtons, respectively, to highlight the HD-tACS modulation to the signals’ variability.

**Figure 2 biomedicines-12-02635-f002:**
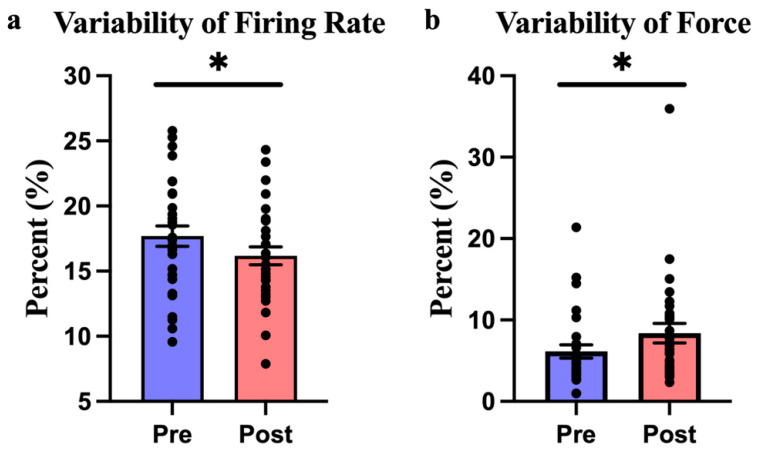
(**a**) Averaged CoV_MUAP_FR values from the hotspot and C3 before and after the HD-tACS intervention. (**b**) Averaged CoV_Force values from the hotspot and C3 before and after the HD-tACS intervention. Data are presented as mean ± SE. A single asterisk (*) indicates a *p*-value < 0.05.

**Figure 3 biomedicines-12-02635-f003:**
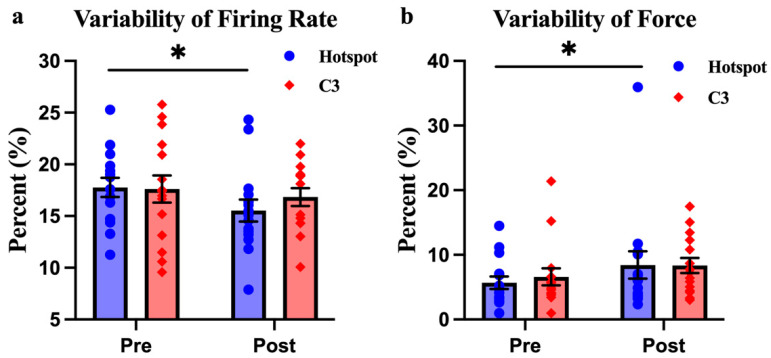
(**a**) CoV_MUAP_FR with HD-tACS applied at hotspot and C3 sites. (**b**) CoV_Force with HD-tACS applied at the Hotspot and C3 sites. Data are presented as mean ± SE. A single asterisk (*) indicates a *p*-value < 0.05.

**Table 1 biomedicines-12-02635-t001:** MUAP characteristics and CoV_Force after HD-tACS (pre-tACS vs. post-tACS). Data are presented as mean (SD).

	Hotspot			C3		
	Pre	Post	%Change	Pre	Post	%Change
# of MUs (n)	18.5 (6.5)	18.7 (6.5)	1.08%	19.7 (4.6)	19.1 (5.0)	−3.05%
MUAP Amplitude (μV)	79.9 (22.3)	81.3 (22.3)	1.75%	86.3 (33.2)	90.8 (33.2)	5.21%
MUAP Firing Rate (Hz)	11.3 (0.9)	11.5 (1.0)	1.77%	11.6 (0.8)	11.6 (0.9)	0.00%
CoV_MUAP Firing Rate (%)	17.8 (3.6)	15.5 (4.1) *	−12.92%	17.6 (5.1)	16.8 (3.4)	−4.55%
CoV_Force (%)	5.7 (3.7)	8.4 (8.1) *	47.37%	6.6 (5.1)	8.3 (4.5)	25.76%

Data are presented as mean ± SE. A single asterisk (*) indicates a *p*-value < 0.05.

## Data Availability

The data supporting the reported results of this study are stored in the lab and can be shared upon reasonable request. Due to privacy and ethical considerations, the data are not publicly available. Researchers interested in accessing the data may contact the corresponding author.

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
