# Peer review of "Efficacy of High-Definition Transcranial Alternating Current Stimulation (HD-tACS) at the M1 Hotspot Versus C3 Site in Modulating Corticospinal Tract Excitability"

_biomedicines, 2024, doi:10.3390/biomedicines12112635_

Round 1

Reviewer 1 Report

Comments and Suggestions for Authors

-Please verify the presence of each keyword within the MeSH browser. 

-Please add the design of the study to the abstract. 

-please also clearly state the design in the methods section. RCT? or?

-Was the study registered to any clinical trial registry? 

-"The study protocol 79 was approved by the Committee for the Protection of Human Participants (CPHS) of the 80 University of Texas Health Science Center at Houston" Please provide protocol number and approval date. 

-Please add conflicts of interest statement. 

-Detailed specifications of all devices and software, including their manufacturers and countries of origin, are required. 

-Results lack any demographic data of the sample. 

Author Response

Please verify the presence of each keyword within the MeSH browser.
REPLY:  Thank you for your suggestion. We have revised the keywords to ensure they align with MeSH terms. The updated keywords are: "Transcranial Alternating Current Stimulation," "Action Potentials," "Cortical Excitability," and "Motor Cortex."

Please add the design of the study to the abstract.
REPLY:  Thank you for this valuable feedback. We have revised the Methods section of the abstract to include the study design:
"Methods: This study used a randomized crossover trial design, where fifteen healthy participants performed a paced ball-squeezing exercise while receiving high-definition transcranial alternating current stimulation (HD-tACS) at 21 Hz and 2 mA for 20 minutes. HD-tACS targeted either the flexor digitorum superficialis (FDS) hotspot or the C3 site, with the order of stimulation randomized for each participant and a 1-week washout period between sessions. Motor unit activities were recorded from the FDS."

Please also clearly state the design in the methods section. RCT? or?
REPLY:  Thank you for your comments. We have clarified the design in the Methods section:
“This study employed a randomized crossover trial design”.

Was the study registered to any clinical trial registry?
REPLY:  Thank you for your comments. The study was not registered in a clinical trial registry because it was an exploratory trial conducted with healthy participants. The primary objective was to investigate the physiological effects of HD-tACS on motor unit activity in a controlled experimental setting, rather than to evaluate a clinical treatment.

"The study protocol was approved by the Committee for the Protection of Human Participants (CPHS) of the University of Texas Health Science Center at Houston." Please provide the protocol number and approval date.
REPLY:  The study protocol number is HSC-MS-22-0046, and the initial approval date was 03/30/2022.

Please add a conflicts of interest statement.
REPLY:  Thank you for this suggestion. We have added the statement as: The authors declare no conflicts of interest. But it is the editor’s decision whether to include this statement in the manuscript.

Detailed specifications of all devices and software, including their manufacturers and countries of origin, are required.
REPLY:  Thank you for this suggestion. We have provided the following details:

Four force transducers (PCB Piezotronics Inc., Depew, NY, USA) were positioned in front of the participant's dominant hand, with spacing adjusted to comfortably accommodate the index, middle, ring, and little fingers. Visual targets were displayed on a screen, programmed using LabVIEW (National Instruments, Austin, TX, USA). The variability of force was assessed by calculating the CoV of Force (CoV_Force), which was the ratio of the standard deviation (SD) of the total force to its mean for the entire 40-second period, multiplied by 100. The total force was the sum of the forces produced by each individual finger. All computations were performed using MATLAB 2022b (MathWorks, Natick, MA, USA). All statistical analyses were performed using SPSS 27 (IBM Corp., Armonk, NY, USA), with the significance level set at p < 0.05.

Results lack any demographic data of the sample.
REPLY: Thank you for pointing this out. We have added the following description to the Results section: "A total of 15 healthy right-handed participants participated in this study (5 females, 29.8 ± 11.9 years of age; 10 males, 34.6 ± 11.2 years of age)."

Reviewer 2 Report

Comments and Suggestions for Authors

This manuscript examines the effects of high-definition tACS at the M1 hotspot versus C3 site on the modulation of motor unit activities.

The manuscript was clearly written. The study is original and relevant for this research field.

I suggest is to add more detail to the results section to make it clearer for readers who are not specialists in tACS.

Also I suggest to define in the figure legends the acronyms employed in the figures.

The authors could include more detailed commentary on the possible causes of outlayers in figure 3.

Author Response

I suggest is to add more detail to the results section to make it clearer for readers who are not specialists in tACS.

REPLY: We appreciate review’s comments, we have added more detail to tACS and the outcome measures.

A total of 15 healthy right-handed participants participated in this study (5 females, 29.8 ± 11.9 years of age; 10 males, 34.6 ± 11.2 years of age). Overall, the application of high-definition transcranial alternating current stimulation (HD-tACS) significantly influenced two key measures: the variability in the motor unit firing rate (CoV_MUAP_FR) and the variability of the force produced by the participants (CoV_Force). Other characteristics related to the motor unit action potential (MUAP) were not significantly affected (Table 1).

For CoV_MUAP_FR (the measure of how consistently the muscle's motor units fire), there was a significant effect over time, meaning that HD-tACS caused changes in the consistency of motor unit firing as the session progressed (F(1,28)=5.012, p=0.033, η²=0.152, Fig 2a). However, there were no significant differences based on the stimulation site (F(1,28)=0.184, p=0.671, η²=0.007) or in the interaction between time and stimulation site (F(1,28)=1.171, p=0.288, η²=0.04). A post hoc analysis showed that these changes were only significant when HD-tACS was applied to the FDS hotspot. Specifically, there was a 12.92% reduction in the variability of the motor unit firing rate after HD-tACS was applied to the FDS hotspot (p = 0.026, Fig. 3a), but no significant change when stimulation was applied to the more general C3 site (p = 0.42).

For CoV_Force (the measure of how consistently the participants generated force), there was also a significant effect over time, indicating that the stimulation affected how consistently force was applied throughout the task (F(1,28) = 5.611, p=0.025, η²=0.167, Fig 2b). However, there were no significant differences based on the stimulation site (F(1,28) = 0.051, p=0.822, η²=0.002) or interaction effects (F(1,28) = 0.286, p=0.597, η²=0.01) effects. A post hoc analysis revealed that the variability of force output increased by 47.37% after HD-tACS was applied to the FDS hotspot (p = 0.049, Fig. 3b), but no significant changes were found at the C3 site.

Also I suggest to define in the figure legends the acronyms employed in the figures.

REPLY: Thank you for your valuable comments. We have made the suggested modifications to the figures, as outlined below.

The authors could include more detailed commentary on the possible causes of outlayers in figure 3.

REPLY: We thank you for the comments. A few outliers are present, especially in the post-intervention measurements for both the hotspot and C3 sites. These outliers may be due to individual differences in muscle control and variability in motor unit recruitment strategies during the force task. Certain participants may have more variable neuromuscular control, leading to higher CoV_Force values after the intervention. This variability could be influenced by factors such as fatigue or individual differences in response to the HD-tACS stimulation. Future studies may consider including larger sample sizes to reduce the impact of individual outliers on the overall analysis.

Reviewer 3 Report

Comments and Suggestions for Authors

The purposes of the study were to determine if HD-tACS could modulate spinal motor neuron behavior through the corticospinal tract using a focal montage and at a high intensity and to determine if HD-tACS at the the upper limb hotspot could modulate the behavior differently from the conventionally C3 scalp site. Fifteen subjects completed the study. The study was a within-subjects design and each subject completed two experiments (HD-tACS over the hotspt, or HD-tACS over the C3 location) with a week washout. HD-surface EMG was taken from the flexor digitorum muscles. tACS was given at 2 mA and at 21 Hz. During the HD-tACS intervention, participants were instructed to gently squeeze a small stress ball at a pace of 15 squeezes per minute until the end of the intervention. The authors predicted that HD-tACS would reduce motor unit firing rate variability and that placing the stimulation over the hotspot would lead to more modulation of motor unit activity. The main findings were generally in line with these expectations. Overall, the study seemed to be conducted carefully, was easy to understand, had a novel design with new methodological tools (HD-tACS, HD-sEMG, and was very well-written with few grammatical or typographical errors. I think the study adds to the literature on the topics of tACS and cortical excitability. The study should be of interest to readers of Biomedicine and researchers is several related fields. 

The study had several strengths and a few weaknesses. Strengths: 1) novel methodology and techniques; 2) well-written; and 3) builds on authors prior research and overall literature. Weaknesses: 1) no SHAM condition and 2) blinding not described.  

I have a few comments of various types that the authors should consider and a few minor corrections to suggest. 

Major:

1. I think the major weakness of the study is that there was no SHAM condition to compare the other conditions to. Thus, placebo effects could not be teased out. The authors acknowledged this limitation, but nonetheless I think more info needs to be given as to why this is not a major issue with the paper.

2. No information on blinding was given, I assume it was single blind ? It appears it was not double blind. This is also potentially an issue that should be explained in more detail by the authors. 

Minor: 

  1. Figure 2 needs to be clarified. I assume that the data has been collapsed across conditions and each point is one observation correct? Why are they not put in blue and red dots like Figure 3? The way it is written in the legend may be confusing to some readers.
  2. Lines 219 to 222. “Specifically, variability in firing rate was reduced by approximately 13% and variability in force by 47% when stimulation was applied over the hotspot, compared to only 5% and 26% reductions, respectively, at the C3 site”  The way this is written does not match Figure 3 or Table 1 results. Force variability was increased for instance in both conditions.
  3. Many times in the paper such as lines 197 to 200 the authors say firing rate or force variability was modulated. This gives the reader less than ideal information many times. Just state the direction of the change every time to lower confusion.
  4. It is very counterintuitive the discharge rate variability would go down but force variability would go up. Can the authors explain this better in the discussion.
  5. Lines 254, Bad transition to synchronization. The argument is also hard to follow and speculative. Synchronization was not measured, I am not sure that increased discharge rate variability automatically means greater synchronization. Other studies not cited by the authors have shown no effect of synchronization of force variability. It is difficult to determine what the authors are trying to get across in this paragraph.
  6. Lack of blinding not mentioned in the limitations.

Author Response

please see attachment as well: 

Major:

  1. I think the major weakness of the study is that there was no SHAM condition to compare the other conditions to. Thus, placebo effects could not be teased out. The authors acknowledged this limitation, but nonetheless I think more info needs to be given as to why this is not a major issue with the paper.

REPLY: We appreciate the reviewer’s comments and fully acknowledge the absence of a sham condition as a limitation. The decision not to include a sham condition was based on several considerations. First, the primary aim of the study was to compare the effects of HD-tACS at two closely related but distinct brain regions. Our main focus was to examine the differences in the neuromodulatory effects between these two locations. Since the study employed a randomized crossover design, participants experienced both interventions, and any placebo effects would likely impact both conditions equally. Furthermore, C3 is a site that is commonly used for tACS or tDCS. The result of no neuromodulatory effects from ACTIVE HD-tACS at C3 suggests that it is not likely to observe a placebo effect in this study. Since there are no effects of HD-tACS at C3 on the corticospinal excitability in our recent study (Meng et al. 2024), this condition could be served as a sham condition. Finally, the following articles with similar objectives also did not include a sham condition. Therefore, we believe that the absence of a sham condition does not pose a significant issue in the current study design.

https://physoc.onlinelibrary.wiley.com/doi/full/10.14814/phy2.14595

https://www.sciencedirect.com/science/article/pii/S0168010220303904

https://www.frontiersin.org/journals/neurology/articles/10.3389/fneur.2022.830976/full

  1. No information on blinding was given, I assume it was single blind ? It appears it was not double blind. This is also potentially an issue that should be explained in more detail by the authors. 

REPLY: Thank you for pointing this out. The study was conducted using a single-blind design, where the participants were unaware of the specific stimulation conditions (HD-tACS at the hotspot or C3), but the investigators administering the interventions were aware of the conditions. We opted for a single-blind design due to the challenges of implementing a double-blind approach in a crossover study where precise stimulation locations were required. Ensuring accurate electrode placement at specific sites (e.g., the motor hotspot) required the involvement of trained personnel who were aware of the condition being applied. However, we do acknowledge that a double-blind design could have further minimized potential bias, and this limitation has been noted in the manuscript.

Minor: 

  1. Figure 2 needs to be clarified. I assume that the data has been collapsed across conditions and each point is one observation correct? Why are they not put in blue and red dots like Figure 3? The way it is written in the legend may be confusing to some readers.

Reply: thanks for pointing this out. We have modified Figure 2 to improve its readability.

  1. Lines 219 to 222. “Specifically, variability in firing rate was reduced by approximately 13% and variability in force by 47% when stimulation was applied over the hotspot, compared to only 5% and 26% reductions, respectively, at the C3 site”  The way this is written does not match Figure 3 or Table 1 results. Force variability was increased for instance in both conditions.

Reply: We appreciated the review’s comments. We have modified the results section, Specifically, variability in firing rate was reduced by approximately 13%, and variability in force increased by 47% when stimulation was applied over the hotspot, compared to only a 5% reduction and a 26% increase, respectively, at the C3 site.

  1. Many times in the paper such as lines 197 to 200 the authors say firing rate or force variability was modulated. This gives the reader less than ideal information many times. Just state the direction of the change every time to lower confusion.

Reply: We thank you for the reviewer’s valuable comments. We have revised the discussion throughout the text to improve clarity and consistency for the readers. See revision for lines 197-200 below:

“Our findings of reduced variability in motor unit firing rate provide further evidence and extend previous findings that tACS can modulate corticospinal excitability at the spinal level. Even though no main effect of Location was reported, additional analysis indicated that the reduction in firing rate variability and the increase in force variability were likely driven by the modulation applied over the hotspot, rather than the C3 site.”

  1. It is very counterintuitive the discharge rate variability would go down but force variability would go up. Can the authors explain this better in the discussion.

REPLY: We appreciated reviewer’s comment: we have added the following discussions to address this phenomena: “The results of the current study demonstrated that the variability of the MUAP firing rate decreased on average by approximately 9% after 20 minutes of tACS treatment, suggesting that firing of motor units became more synchronized. However, the variability of isometric contraction force increased by approximately 36%. This seems to be counter-intuitive at the first look. It does actually reflect underlying mechanisms of neuromuscular force control. Before HD-tACS treatment, motor units fire and oscillate at certain frequencies with individual peaks and troughs. When firing of motor units becomes less variable after HD-tACS treatment, more oscillation peaks of motor unit firing are aligned or get closer. Such synchrony of oscillation peaks increases force output (Taylor et al. 2003). Similarly, synchrony of oscillation troughs decreases force output. In other words, increased variability of total finger force is a behavioral reflection of decreased variability of motor unit firing. Given there were no changes in the number of firing motor units and their amplitude and firing rate, the total force remains unchanged. This phenomenon is visually displayed in Figure 1..”

Lines 254, Bad transition to synchronization. The argument is also hard to follow and speculative. Synchronization was not measured, I am not sure that increased discharge rate variability automatically means greater synchronization. Other studies not cited by the authors have shown no effect of synchronization of force variability. It is difficult to determine what the authors are trying to get across in this paragraph.

REPLY: We appreciate the feedback and have removed the paragraph. Additionally, we have expanded the discussion on the variability of firing rates and force.

  1. Lack of blinding not mentioned in the limitations.

REPLY: We appreciate the reviewer’s comments. We have now included this point in the limitations section. Specifically, a limitation of this study is the absence of double-blinding. While participants were unaware of the specific intervention (single-blind design), the investigators knew the stimulation site. This lack of blinding could introduce potential bias, particularly from the investigators, which may influence the application of the intervention or the interpretation of results.

Round 2

Reviewer 3 Report

Comments and Suggestions for Authors

The authors appear to have adequately addressed all of my prior comments.